# Simulation and Improvement of Patients’ Workflow in Heart Clinics during COVID-19 Pandemic Using Timed Coloured Petri Nets

**DOI:** 10.3390/ijerph17228577

**Published:** 2020-11-19

**Authors:** Masoomeh Zeinalnezhad, Abdoulmohammad Gholamzadeh Chofreh, Feybi Ariani Goni, Jiří Jaromír Klemeš, Emelia Sari

**Affiliations:** 1Department of Industrial Engineering, West Tehran Branch, Islamic Azad University, Tehran 1468763785, Iran; 2Sustainable Process Integration Laboratory–SPIL, NETME Centre, Faculty of Mechanical Engineering, Brno University of Technology, VUT Brno, Technická 2896/2, 61669 Brno, Czech Republic; chofreh@fme.vutbr.cz (A.G.C.); goni@fme.vutbr.cz (F.A.G.); jiri.klemes@vutbr.cz (J.J.K.); 3Department of Management, Faculty of Business and Management, Brno University of Technology, VUT Brno, Kolejni 2906/4, 61200 Brno, Czech Republic; 4Department of Industrial Engineering, Faculty of Industrial Technology, Universitas Trisakti, Kyai Tapa No 1, Grogol, Jakarta Barat 11440, Indonesia; emelia@trisakti.ac.id

**Keywords:** healthcare systems, hospital, COVID-19, discrete-event simulation, waiting time, timed coloured Petri net, heart clinic

## Abstract

The COVID-19 epidemic has spread across the world within months and creates multiple challenges for healthcare providers. Patients with cardiovascular disease represent a vulnerable population when suffering from COVID-19. Most hospitals have been facing difficulties in the treatment of COVID-19 patients, and there is a need to minimise patient flow time so that staff health is less endangered, and more patients can be treated. This article shows how to use simulation techniques to prepare hospitals for a virus outbreak. The initial simulation of the current processes of the heart clinic first identified the bottlenecks. It confirmed that the current workflow is not optimal for COVID-19 patients; therefore, to reduce waiting time, three optimisation scenarios are proposed. In the best situation, the discrete-event simulation of the second scenario led to a 62.3% reduction in patient waiting time. This is one of the few studies that show how hospitals can use workflow modelling using timed coloured Petri nets to manage healthcare systems in practice. This technique would be valuable in these challenging times as the health of staff, and other patients are at risk from the nosocomial transmission.

## 1. Introduction

COVID-19 pandemic has rapidly spread across the world [1]. It forced health services to impose instant restrictions on every type of hospital activity, to shield patients and to prepare for emergencies [2]. This circumstance happened mainly in the early phase of the epidemic, where the majority of patients scheduled for examination were informed that they had been postponed indefinitely [2]. This problem causes dissatisfaction and discomfort. In this challenging time, there is an urgent need to quickly adapt and apply new processes and revise standard care models, technologies, and workflow [3]. As the pandemic expands, the authorities are challenged to develop or keep the health status of their hospitals [3].

Sanitary and healthcare systems have several complications at all levels [4]. System complications and interactions make it difficult even to predict their performance [5]. In this case, the researchers are often accustomed to queue theory [6], simulation, and Petri nets [7] to describe the structure of the healthcare system. The motion of the patient demonstrates the healthcare system’s ability to provide efficient, reliable, and quick service [3].

Thousands of patients each day face several hours of delay in the treatment process [6]. In mitigating this problem, Petri nets are an appropriate tool to analyse and improve the performance of the healthcare system [8] as there are various uncertainties in the outpatient healthcare system (e.g., the time to examine is unpredictable and usually differs in each case). Outpatient waiting time is another significant factor in patient satisfaction [7]. Reducing patient waiting times can lower associated costs and increase the efficiency of the service provision system. One approach to examining and promoting the service provision process is to analyse patient pathways [6], which this study attempted to investigate by optimising patient waiting times and analysing the significant factors in this problem.

With the ever-increasing advancement in technology and rapid organisational growth, one of the main concerns would be high-quality service and customer absorption [7]. Researchers and scientists have conducted extensive studies and offered novel methods to achieve this important objective. One important factor that exists in all the algorithms and techniques offered is the waiting time [9].

Conventional and modern systems strive to satisfy their customers in a competitive atmosphere [8]. To achieve this goal successfully, they should identify working bottlenecks and provide solutions for them. Since in conventional systems, expenditures incurred do not comply with new requirements [9], new approaches should be offered to meet the updated requirements before the new architecture is implemented.

Possibilities should be considered in the design and development of novel systems [10] so that the capability of the new architectures and methods can be evaluated at the least cost. The development of a more appropriate system can be done by modelling the patterns of the system architecture. Pattern selection, modelling, and evaluation are an important decision that should be emphasised [10]. The developer can design the architecture using modelling before allocating additional resources and returning to the pre-design step to ensure a reliable system.

Modern and service-based systems play a crucial role in financial and economical provision across countries [11]. World experiences show that conventional systems are not dynamic and stable [12]. Based on this assumption, practitioners look for innovations so that they can increase productivity and provide better services to their customers.

Innovation often emphasises the organisational structure [13]. However, methods that emphasise decreasing queue length and estimating customer needs in terms of new technology are considered to be more prosperous than other competitors [14]. One method used to assess waiting time is to model the workflow of service-based organisations whose functions are evaluated in different situations [13].

COVID-19 is genuinely a global crisis that needs to be immediately resolved as it causes various problems and forces all parties to innovate in various fields to solve them immediately [15]. In the healthcare field, there is a need to achieve optimal waiting times in clinical workflow and identify and resolve bottlenecks [2]. The COVID-19 pandemic forces clinics to modify their workflows for urgent patients [16]. Particularly, this virus has posed many challenges for heart clinics. Patients with cardiovascular disease represent a vulnerable population when suffering from COVID-19 [17]. As warned by Fersia et al. [18], the provision of cardiac services was severely restricted due to a shift in focus in dealing with the surge of patients with COVID-19 and patients’ reluctance to seek medical help during the lockdown period. It is, therefore expected that there will be another surge of patients seeking cardiology care and that services need to plan to treat these patients early and urgently to prevent any long-term complications [18]. This problem motivates this study to propose changes that should be made in the workflows of heart clinics to avoid a collapse of the healthcare system. An executable model is offered using timed coloured Petri nets to demonstrate the waiting time. This article provides a solution for reducing patient waiting time through workflow modelling and simulation. Simulations to determine the current status and to develop a novel process are carried out through discrete-event simulations using timed coloured Petri nets.

The arrangement of this article is as follows. Section 2 relates to the analysis of the reviewed literature, which concludes with the gap identification. Section 3 presents the research methodology, and the next section presents the results of data analysis. In Section 5, details related to the planned scenarios are discussed and explained. The last section concludes the study and proposes a few potential research recommendations.

## 2. Literature Review

Several studies have investigated various problems that occur in hospitals caused by Coronavirus. In Italy, Bettinelli et al. [2] studied the workflow of an orthopaedic clinic during the COVID-19 pandemic. They summarised all the changes that had to be made to prevent the healthcare system’s downfall in the most affected areas and provided an effective flowchart. They proposed a model that shows the workflow for patients arrived in the emergency room (ER) in an Orthopaedic Hub during the coronavirus disease emergency, as given in Figure 1. In the model, the hub and spoke organisation was enforced by an immediate-effect regional decree.

In another study, Wei et al. [19] improved the workflow of radiotherapy procedures during the COVID-19 pandemic in a cancer hospital in Wuhan. They affirmed that a stringent COVID-19 screening protocol was implemented at their centre, and the workflow of radiotherapy was optimised for combating the epidemic.

Simulation techniques have been used in several hospital-related studies. This technique makes a safe analytical lens into the process; therefore, flow can be optimised, and risk minimised [20]. Das [21] studied the effect of the COVID-19 outbreak on the workflow of an endoscopy centre. This study developed a discrete event simulation-based model to measure the impact of the changes on the performance indicators related to COVID-19-related workflow and cost per case compared with the pre-COVID-19 baseline. The results show that the post-COVID-19 suggested workflow changes have a significant impact on productivity and operational metrics and, in turn, adversely impact financial indicators. There has been a substantial reduction in staff utilisation resulting in a growth in total patient waiting time, facility time, and cost per case due to a bottleneck caused by pre-procedure COVID-19 testing and screening while practising.

Diaz and Dawson [20] used simulation to develop a COVID-19 resuscitation procedure in the emergency department of paediatrics. They concluded that simulation might be used to formulate COVID-19 spaces, processes, and workflows. A brief overview of studies published in the year 2020 relating to investigations of changes made by COVID-19 to patient workflows is presented in Table 1.

Table 1 shows that many studies have highlighted the importance of efficient changes in hospital processes in 2020. Various specialised clinics including radiology, surgery, heart, ICU, emergency, endoscopy and radiology, oncology, radiotherapy, and orthopaedic have studied to examine the possibility of improving their work processes to deal with the virus pandemic crisis. However, more than half of the studied papers are just reviews of the relevant literature and do not provide a practical model. For example, in heart clinics, Harjai et al. [16] surveyed the physicians and provided viewpoints from mangers to study the improvement possibility of patient workflow, theoretically. Similarly, Virani et al. [3] reviewed the activities performed for heart failure care during the COVID-19 outbreak and provided viewpoints from leadership within the Canadian Heart Failure Society.

Although publicity about the COVID-19 pandemic has increased recently, the number of studies at heart clinics is considerably lower than at other centres. This virus has posed many challenges for heart clinics, and there is a need to optimise patient flow time so that staff health is less endangered, and more patients can be treated.

The literature review confirms that simulation techniques can be used to model the current processes of treatment in the hospital and identify bottlenecks. However, from the content and methodology perspective, the use of Petri nets in modelling the current situation and implementing improvement scenarios has not been considered in practice. This paper aims to show how a heart clinic can use workflow modelling using timed coloured Petri nets to manage the healthcare system in practice.

## 3. Research Methodology

A brief overview of the research methodology is shown in Figure 2. In the first stage, an initial simulation to evaluate the current workflow of patients in a heart clinic of a private hospital in Tehran was carried out. This activity identified bottlenecks and highlighted long waiting times.

In the second stage, regarding the dramatic increase in the number of patients during the COVID-19 pandemic, three scenarios were proposed to achieve an optimum waiting time. They were simulated, and the best one was selected. The following subsections explain the waiting time modelling and optimisation techniques. Timed coloured Petri net as a simulation tool is described. Workflow modelling and the sequence diagram evaluation, as well as the data collection method, are discussed in the following sub-sections.

### 3.1. Timed Coloured Petri Net

Petri nets show a clear and graphical presentation of the system, along with a mathematical approach [7]. They demonstrate communicative patterns, control patterns, and information flow [6]. Petri nets are considered as a framework to analyse, verify, and evaluate efficiency [7]. Figure 3 illustrates an example of a coloured Petri net. The circles (p*i*) are entitled places that designate the positions of the system. The rectangles (t*i*) are named transitions that represent the activities. The arrows are called arcs, which determine the process of changing the state of the coloured Petri nets when the transition occurs. Each place has a set of indicators called tokens, which carry data values. This condition is different from low-level Petri nets modelling [27].

A coloured Petri net is a tuple N = (P, T, A, Σ, C, N, E, G, I) where P is a set of places, T is a set of transitions, and A is a set of arcs. Sets of places, transitions, and arcs are pairwise disjoint P ∩ T = P ∩ A = T ∩ A = ∅. Σ refers to a set of colours, and C refers to a colour function. It maps places in P into colours in Σ. N is known as a node function. It maps A into (P × T) ∪ (T × P). E refers to an arc expression function. It maps each arc a ∈ A into the expression e. G is a guard function. It maps each transition t ∈ T to a guard expression g. The output of the guard expression should be calculated to Boolean value: true or false. I is an initialisation function [27].

Petri nets are based on situations rather than on events and offer a structural model and behaviour of a concrete-event system. They present a powerful official model based on mathematical structure. The coloured Petri nets are introduced as a developed model of Petri nets. They present a perfect model of a complicated non-synchronisation process system [27].

This network is a two-part graph consisting of two elements: place and transition [11]. Pieces are introduced as colours, guards, and phrases to these places and transitions [7]. In this network, numerical data are carried by pieces (beads). The coloured Petri nets present a perfect model of a complicated non-synchronisation process system [11]. Unlike the Petri nets, the beads (pieces) in this network are unique because each piece has a colour.

### 3.2. Workflow Modelling

Waiting time presents a level of accessibility to system services that can be measured and improved [28]. Long waiting time in certain parts of the system is an obstacle to offering high-quality service, and this could lead to wasted time and customer dissatisfaction [27]. Modelling and evaluation can be used to improve waiting time based on system bottlenecks and using techniques such as workflows and new arrangements [28]. Modelling represents an executable illustration of system characteristics and their behaviour at the time of execution [12]. Various models exist to serve an executable system [6]. This information cannot be modelled to Unified Modelling Language (UML) as the real-world information is mainly uncertain [27]. To solve this problem, a Petri nets model is introduced to describe and execute the UML characteristics [29].

UML is a virtual and standard language to describe the architecture, and it cannot directly evaluate the system [27]. A conversion from the real model to the official one is required, which means that an executable version of the system should be presented to evaluate the system. An executable version of the system is a formal description of the system, which can help to observe and examine the final behaviour of the system before its implementation [30]. Integrated modelling language diagrams are used to illustrate important workflows [27]. An evaluation of the quality characteristics is needed at the system level, along with an executable version of the system.

In this research, the current workflow of a heart clinic was studied. In a simple shape, it has been illustrated in Figure 4.

As shown in Figure 4, after the patient enters the clinic, he/she is admitted and receives an appointment. For a cardiac examination, electrocardiography (ECG) is necessary first, so patients must have an ECG by a technician before the visit. He/she is then examined by a doctor, who must perform an echocardiogram if there are any symptoms. Otherwise, the patient can be logged out. After performing the echocardiography with the doctor’s discretion, the patient should have a sports test and if he/she does not have some symptoms, can leave the system, otherwise will be transferred to the angiography department, and other tests will be performed on him/her.

### 3.3. Sequence Diagram Evaluation

Petri net-based model examines the workflow in the system for different cases [31]. This model assesses the work bottleneck considering the average waiting time [32]. After identifying the bottleneck using the Petri net technique and mapping by model, an optimum waiting time is achieved using trial and error [33].

This study examined the patient workflow in a heart clinic to illustrate the executable method and its verification. The sequence diagram of the heart clinic is presented in Figure 5. It is in line with Figure 4.

As shown in Figure 5, patient reception ward refers the patient to the electrocardiogram ward. The patient visits a practitioner. The practitioner either discharges the patients or refers them to an echocardiogram. There are two cases after an echocardiogram that either discharge the patient or refer him/her to a sports test. Following that, two cases either discharge the patient or refer him/her to angiography. Based on the angiographic result, the patient is eventually hospitalised or discharged.

Given the applications of Petri nets in queue systems, a network is offered for the patient flow. Timed coloured Petri net is used for modelling because the data are not formed homogeneously. An important advantage of this model is its dynamics, which means that the results can be obtained under certain conditions. For simulation, “CPN Tools” software was applied. In this software, “tokens” that contain data are called “values”. “Places” represent the type of data stored, and “arcs” indicate the types of data used and produced. When the entry tokens are the same type, a coloured Petri net has to be used. The place is a workstation where patients enter and start an activity. Its activity is indicated by transactions that exit the input place.

In “CPN Tools”, every place that the patient is referred to and the place where he/she waits are displayed in the same colour. Tokens are determined based on a place. If a specific place only contains patients, the patient token should be allocated. In other places, there may be patients and technicians.

### 3.4. Data Collection

The population of this study includes patients referred to the heart clinic in a private hospital in Tehran. One hundred patients were randomly selected and studied carefully. The data collection was conducted using timing, on different days and hours of the month. The moment patients enter each ward, the moment patients exit each ward, patient flow time, as well as the duration of patient service in each ward were measured and recorded. Based on the data analysis, it was concluded that probability density functions (PDFs) related to the service duration of all wards could be assumed as a “continuous uniform distribution”. The notation for the uniform distribution is X ~ U (a, b), where a = the lowest value of x and b = the highest value of x. The specifications of the wards are summarised in Table 2.

The outcomes of simulation and the results of the analysis are discussed in the subsequent section.

## 4. Results

This section discusses the current state and results of the initial simulation and optimisation simulations to develop a new process.

### 4.1. Initial Simulation

Descriptions of the system workflow are explained using a sequence diagram, presented in Figure 5. Based on this diagram, the executable model of the patient workflow using timed coloured Petri nets in “CPN Tools” software is developed in Figure 6.

By applying the sequence diagram conversion algorithm to the Petri nets in Figure 6, the selection structure is displayed as a transition. The aim is to identify workflow bottlenecks. After identifying the bottlenecks and getting the waiting time, these bottlenecks are resolved using a dynamic model.

Figure 6 shows the status of the current system. The simulation results indicate that the average waiting time for each ward is, 120, 44, 30, 0, 22, 0, 0, and 47 min. At the current situation, a patient should wait for about 4.4 h to get various services. As shown in Figure 7, the main bottleneck in this system is ward A1, namely, “Patient reception”.

As seen in Figure 7, it is obvious that the focus of research should be on reducing the waiting time related to “Patient reception”. Currently, the total waiting time for each patient is 263 min, of which 120 min (46%) belongs to the ward A1.

### 4.2. Scenario Planning and Simulation

The results shown in Figure 7 reveal that there is a bottleneck, namely A1, each patient, on average, spends 120 min waiting there. Since A1 has the highest number of visits, and the patient visits this ward several times during their stay in the system, it has the longest waiting time. There should be a procedure to reduce the patients waiting time and increase their satisfaction. In this regard, several suggestions are presented as follows:changing the patient’s workflow,increasing the number of stafftemporary usage of free human resources of other wards (sharing), andbuying new equipment.

According to the hospital policies and management strategies, as well as regarding limited space available to install new devices and budget constraints, three scenarios were planned to reduce the waiting time in A1. These scenarios are introduced in Table 3.

At the current situation, there is one staff in the patient reception. According to Table 3, in the first and second scenarios, there are three and four staff at A1, and free staff can be used for other wards, wherever they are required. Based on the third scenario, there are only five staff in A1 without sharing free human resources.

#### 4.2.1. The Results of Scenario 1

As mentioned in Table 3, based on the first scenario, there are three staff at A1, and free staff could be shared for other wards, wherever they are needed. The simulation results of this scenario are shown in Figure 8.

Based on Figure 8, the total waiting time in scenario 1 was 192 min and waiting time-related to A1 was halved.

#### 4.2.2. The Results of Scenario 2

In scenario 2, there are four staff at A1, and free staff could be used for other wards, wherever they are required. The simulation results of the second scenario are presented in Figure 9.

Figure 9 shows the total waiting time in scenario 2 was 99 min for each patient. Moreover, compared to the current system, waiting time-related to A1 was reduced from 120 to 8 min.

#### 4.2.3. The Results of Scenario 3

The last scenario assumed there are five staff at A1, and using the free staff of other wards is not allowed. The simulation results of scenario 3 are summarised in Figure 10.

Based on the simulation results of scenario 3, there was no waiting time in the patient reception ward. However, the total waiting time in this scenario was 250 min. Comparing the current situation, overall, little improvement was made in this scenario.

### 4.3. Comparing the Simulation Results

Figure 11 is presented to show the results of the implementation of the proposed scenarios and compare them with the current state of the system.

Comparison of the results of scenarios simulation with the current waiting times is made in Figure 11. Total waiting time related to the current situation and scenarios 1 to 3, respectively, were calculated as 263, 192, 99, and 250 min. It was confirmed that by applying scenario 1, a 27% reduction in total waiting time happened compared to that in the current system. While the implementation of scenario 2 led to a 62.3% decrease in total waiting time, scenario 4 resulted in only a 5% reduction.

Our simulation confirmed that regarding the management policies [34], cost, and other limitations [35], employing three more staff in “patient reception” ward as well as allocating free human resources of other wards to busy ones was the best scenario.

As shown in Figure 11, the results of the third scenario showed that by increasing the number of staff to five without sharing free resources of the other wards, the total waiting time did not change significantly even though the waiting time at A1 was zero. With the increase in the number of staff in the patient reception from 1 to 5, the queues formed in this ward would disappear. However, due to the limited number of equipment, doctors, and technicians, longer queues will be created in other wards.

### 4.4. Framework Development

A framework is proposed in Figure 12 regarding the permanent increase in the number of coronary and cardiovascular sufferers at risk of mortality. This model could be used as a prerequisite for sequence diagrams, as developed in Figure 5, for the management of heart clinics during the COVID-19 outbreak. Flowcharts help make processes easier to understand [36] and speed up the process of activity [37]. Figure 12 illustrates how to separate and direct emergency patients or coronary patients before starting the treatment process so that issues related to the examination and diagnosis of their heart problems can be done in a short time. In this framework, this study attempts to consider the safety of non-coronary patients and staff by emphasising the use of personal protective equipment (PPE) and the necessary information. Two separate work shifts, namely day shift for non-coronary patients and night shift for coronary patients, were considered in this model.

In scenario 2, the best scenario with four reception staff, there are two separate “patient reception” units, to reduce patient flow time in the system and isolate coronary patients as much as possible. It is essential that all staff and patients carefully follow all the COVID-19 prevention guidelines and at the end of each work shift, or at least at the end of the night shift, all sections and equipment are disinfected.

## 5. Discussion

As patients with cardiovascular illness represent a vulnerable population when suffering from COVID-19, there is a need to minimise patient flow time. In this paper, through planning three applicable scenarios, we tried to reduce patient waiting time as much as possible to safeguard staff health and treat more patients safely. Considering the policies of hospital managers as well as cost and space constraints, based on the second scenario, Figure 9, hiring three more staff in “patient reception” ward and allocating free human resources of other wards to busy ones, leads to a 62.3% decrease in total waiting time.

The initial simulation of the current workflow indicates that the average waiting time for each ward includes, 120 min for “Patient reception”, 44 min for “Electrocardiography”, 30 min for “Check-up”, 0 min for “Echocardiography”, 22 min for “Sport test”, 0 min for “Angiography”, 0 min for “Hospitalization”, and 47 min for “Log out”. While the simulation results of the second scenario implementation show, the waiting time related to those wards changed to 8, 24, 20, 6, 8, 7, 3, and 23 min, respectively.

Our research confirms that simulation using Petri nets can be used in modelling the current processes of treatment and implementing improvement. Timed coloured Petri nets are based on situations rather than on events and offer a structural model and behaviour of a concrete-event system.

The framework for heart clinics is generally developed to guide decision-makers in managing the patients during COVID-19 pandemic. It is an essential prerequisite for heart clinics sequence diagrams to separate and direct emergency patients or coronary patients before starting the treatment process. Based on the proposed framework, presented in Figure 12, the examination and diagnosis of patients’ heart problems can be made in a safe mode and in a short time. A separation and isolation strategy must exist to protect other patients at risk from the nosocomial transmission. Hence, in the proposed framework, two separate work shifts, namely day shift for non-coronary patients and night shift for coronary patients, are considered. The framework can be useful for healthcare managers to grasp the essential aspects of developing improvement strategies for staff and patient satisfaction. The COVID-19 pandemic enforces clinics to modify their workflows for urgent patients, so the proposed framework would help them to re-analysis the strategies and actions requiring changes to attain better performance and competitive advantage.

In the absence of cost and space limitations and inconsistencies with hospital management policies and staff resistance to participation, stronger recovery scenarios can be designed, and comprehensive improvements made.

## 6. Conclusions

As coronavirus disease outspreads across the world, hospitals has to prepare for the challenges related to this outbreak. Reforming workflows for quick diagnosis and isolation as well as infection control programs is vital to patients with COVID-19 and to staff and other patients who are at risk from the nosocomial transmission. This study proposed a method using modelling and evaluation of the workflow to achieve an optimum waiting time using timed coloured Petri nets. The heart clinic workflow was described using a unified modelling language and sequence diagram. A sequence diagram that represents the system behaviour was converted to an executable Petri network. The bottlenecks whose waiting times were longer were identified, and following this identification, the optimised waiting time was achieved by running the planned scenarios. One of the advantages of this proposed method compared to other methods is taking advantage of the dynamic model and timed coloured Petri nets. In conclusion, simulation is regarded as a modelling tool to evaluate the processes of hospitals, find their problems, and provide improvement solutions that lead to profitability and value creation. Findings suggest how many advantages are detectable if Petri nets were used for simulation and workflow modelling.

Forthcoming studies may focus on other criteria, such as reliability, and might consider the post-COVID-19 era and its challenges. Other specialised clinics could be studied with the proposed research methodology in this article, taking into account their specific features and characteristics. Although this study was conducted in Iran, since the treatment of heart patients is usually similar according to the standards of the medical community, the results can be used by other heart clinics with a few changes.

## Figures and Tables

**Figure 1 ijerph-17-08577-f001:**
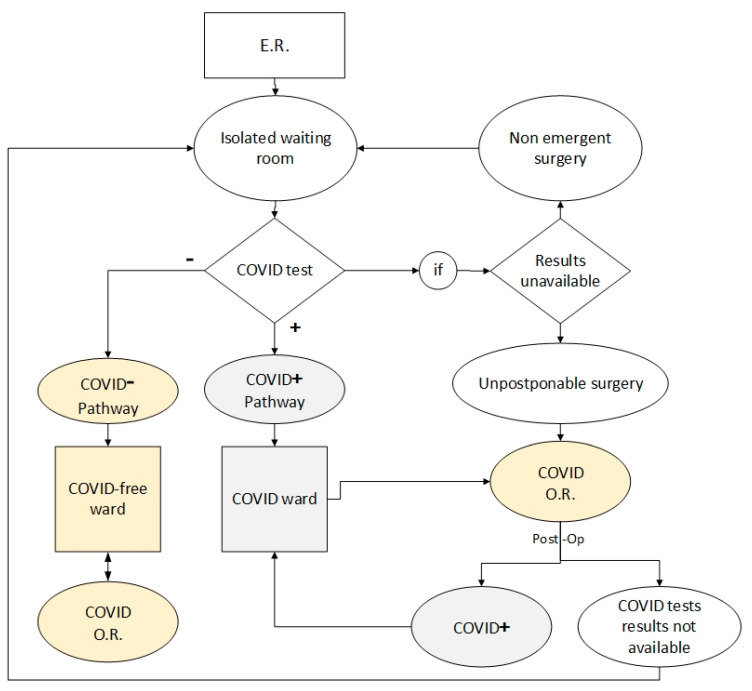
Workflow for orthopaedic patients, amended from [2].

**Figure 2 ijerph-17-08577-f002:**
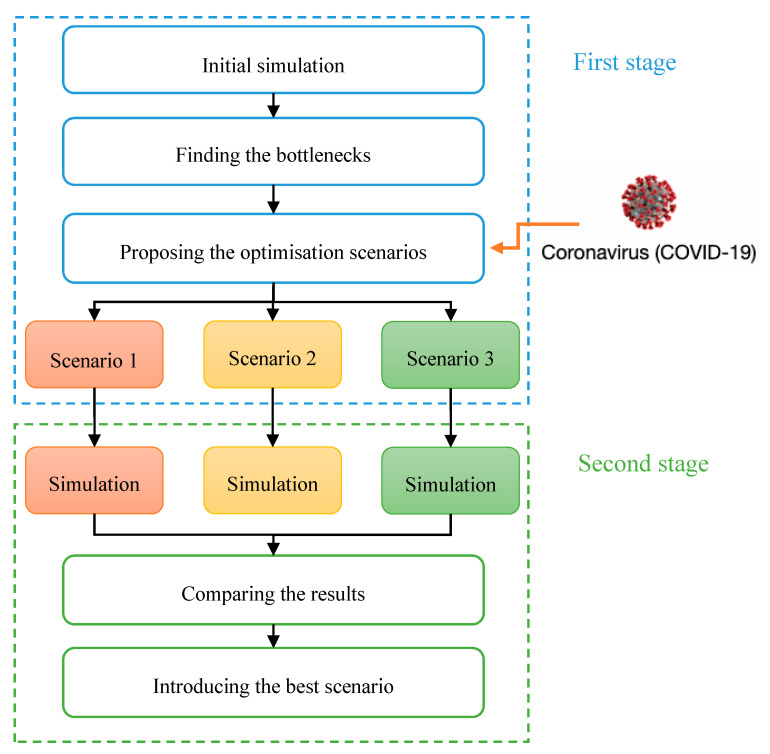
The research methodology.

**Figure 3 ijerph-17-08577-f003:**
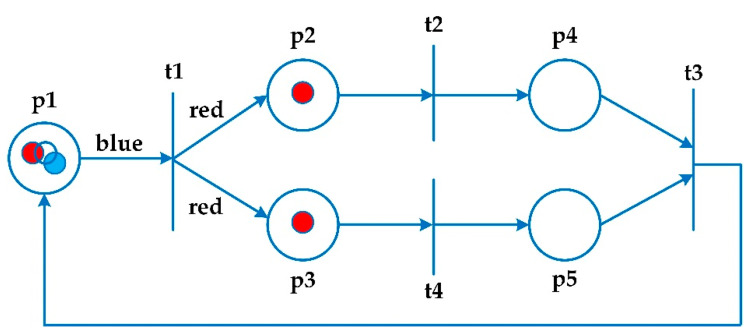
A sample coloured Petri net diagram, amended from [27].

**Figure 4 ijerph-17-08577-f004:**
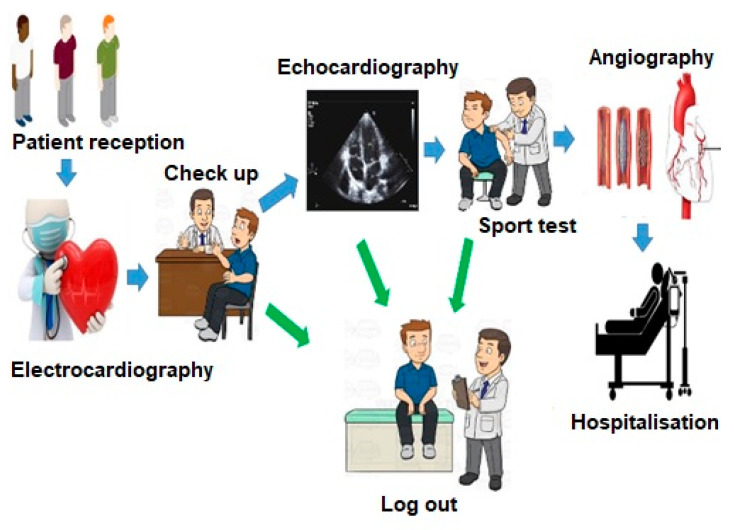
A simple schematic of patient workflow in a heart clinic.

**Figure 5 ijerph-17-08577-f005:**
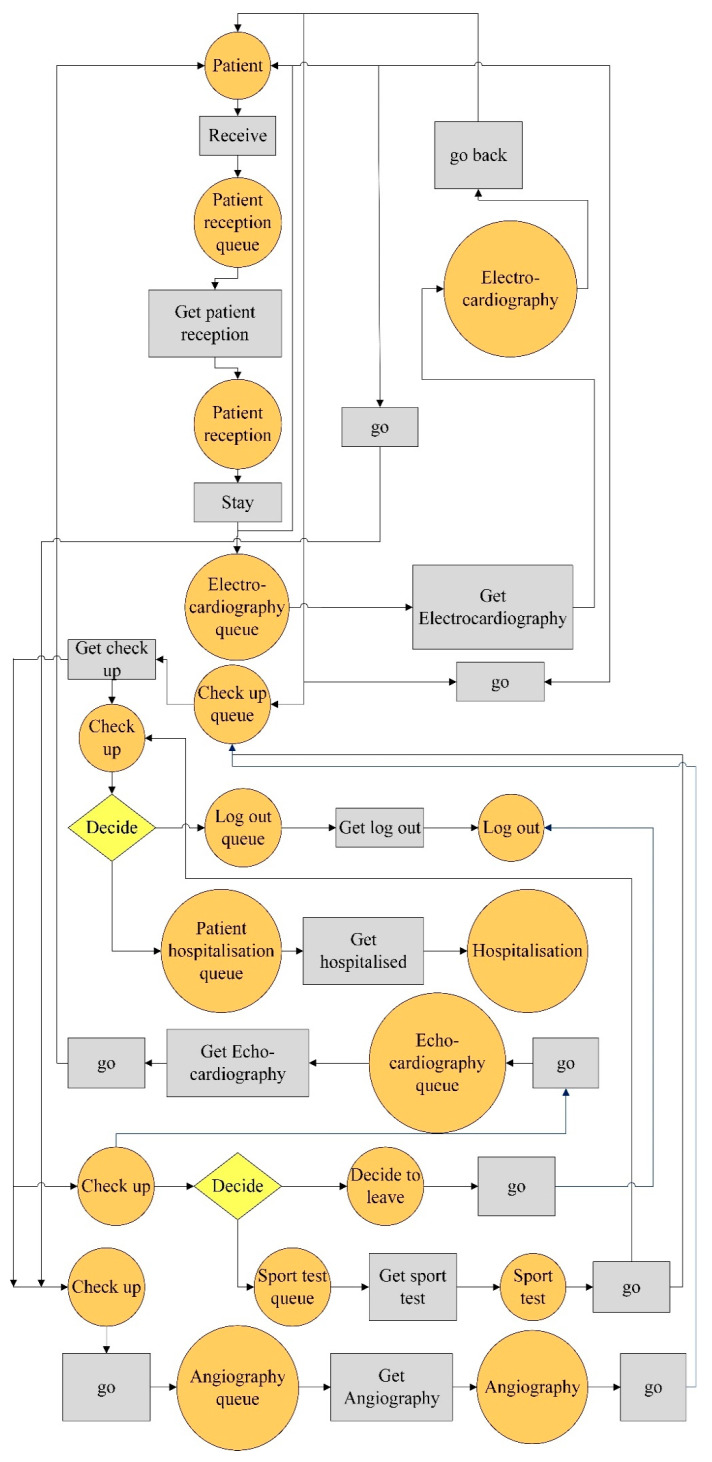
The sequence diagram of the heart clinic.

**Figure 6 ijerph-17-08577-f006:**
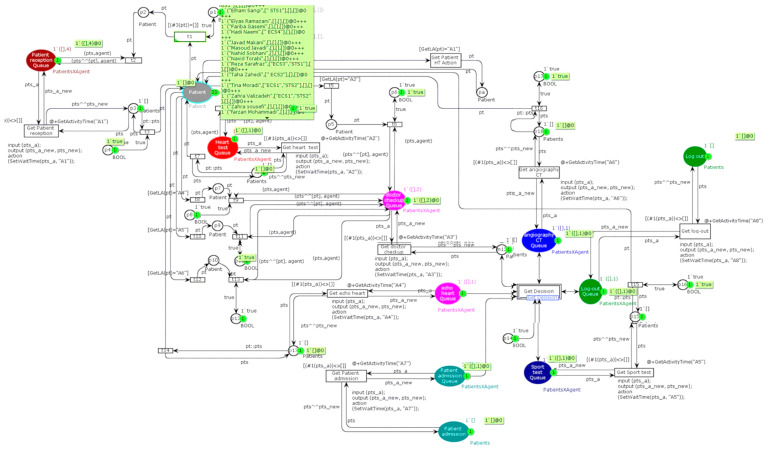
The Petri model of patient’s workflow.

**Figure 7 ijerph-17-08577-f007:**
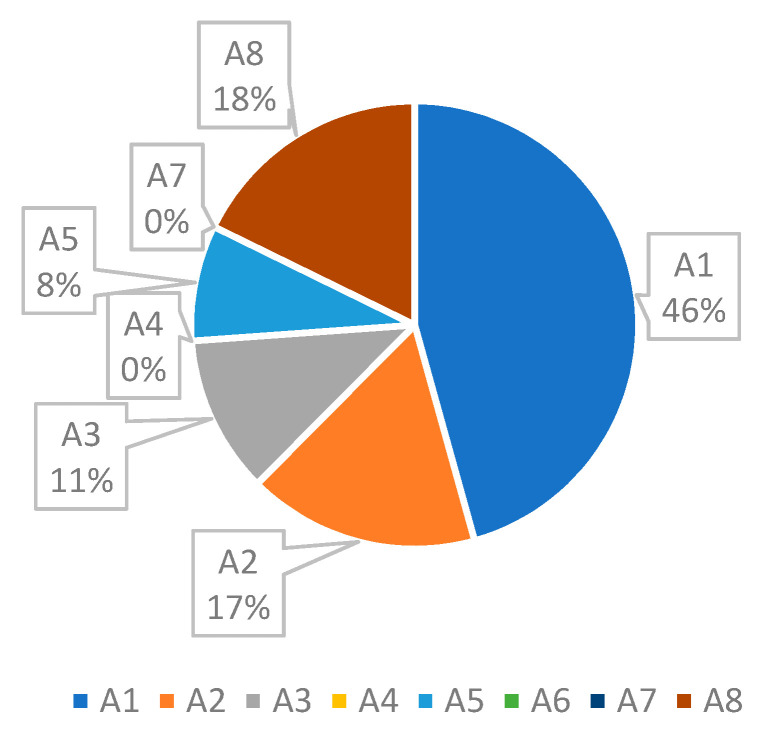
Waiting time in various wards of the heart clinic (current situation).

**Figure 8 ijerph-17-08577-f008:**
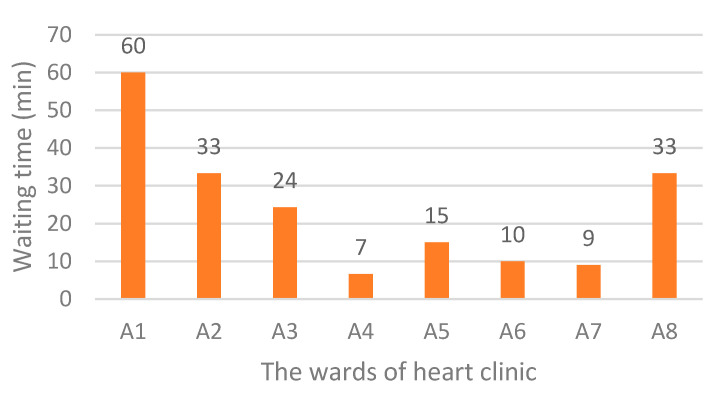
Waiting times after the first scenario implementation.

**Figure 9 ijerph-17-08577-f009:**
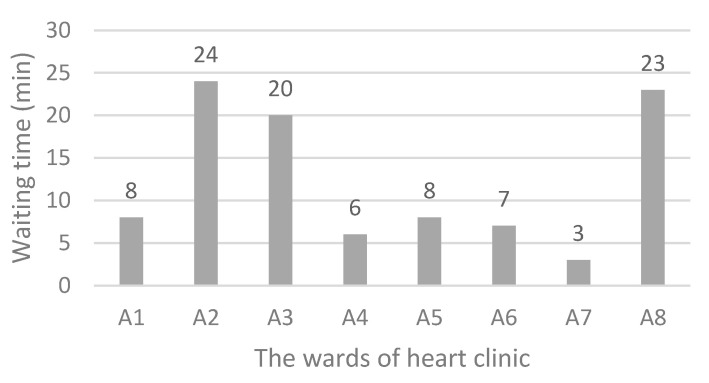
Waiting times after the second scenario implementation.

**Figure 10 ijerph-17-08577-f010:**
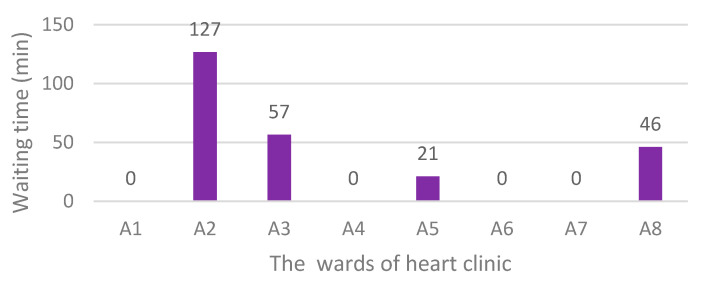
Waiting times after the third scenario implementation.

**Figure 11 ijerph-17-08577-f011:**
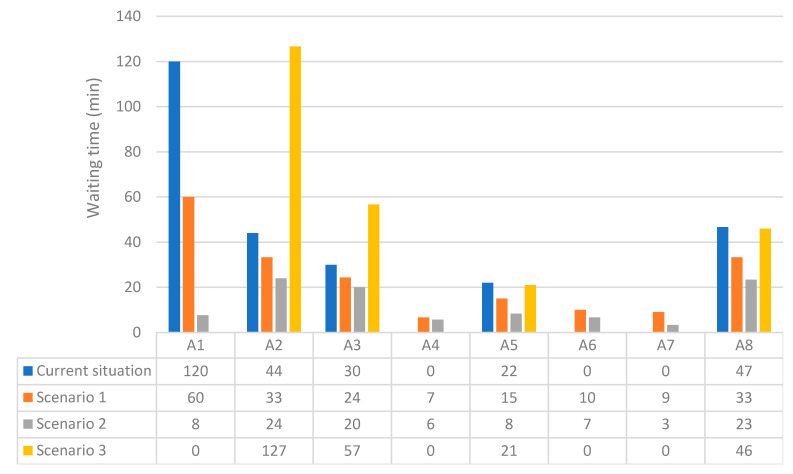
Comparison of scenarios simulation results with the current waiting times.

**Figure 12 ijerph-17-08577-f012:**
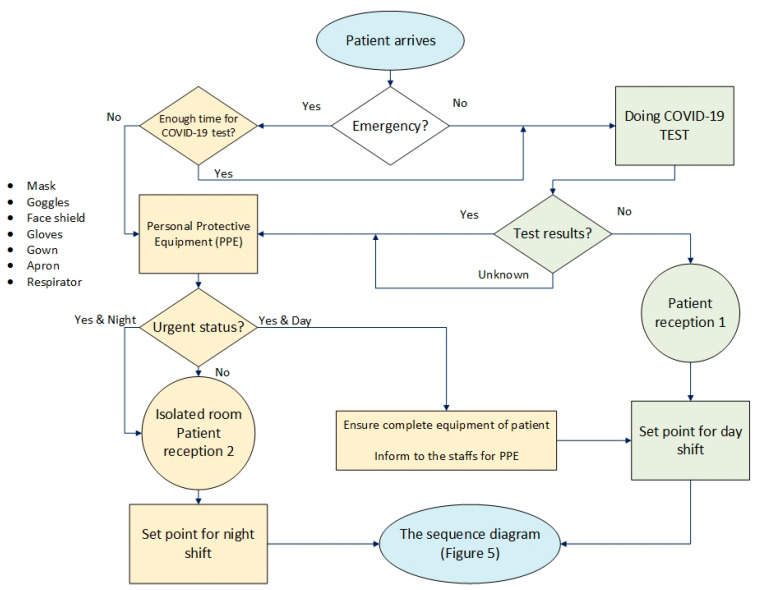
The proposed framework for heart clinics during COVID-19 pandemic.

**Table 1 ijerph-17-08577-t001:** Recent research regarding changes in patient workflow during the COVID-19 pandemic.

Author	Objective	Specialised Clinic	Research Method	Research Highlights
Virani et al. [3]	Optimising access to heart failure care during the COVID-19 outbreak	Heart	Review of the activities performed, and the experiences gained	The study provided viewpoints from leadership within the Canadian Heart Failure Society.
Quraishi et al. [22]	Study of the changes related to the off-site radiology workflow due to the COVID-19 pandemic	Radiology	A survey of 174 radiologists used frequency and descriptive statistics to perform χ^2^ analyses and nonparametric Mann–Whitney *U* tests.	The bulk of radiology practices have leveraged internal teleradiology for regular workday shifts and found an adequate benefit to consider continuing internal teleradiology after the epidemic passes.
Dexter et al. [23]	Proposing strategies for daily operating room management following resolution of the critical phase of the coronavirus pandemic	Ambulatory surgery	A narrative review	The economic costs of the COVID-19 pandemic for ambulatory surgery centres were identified.
Harjai et al. [16]	Providing a road map for clinicians and healthcare delivery systems	Heart	Surveyed 16 physicians across three hospitals.	Most follow-up visits can be done via telemedicine rather than in-person visits.
Phua et al. [24]	Providing recommendations for serious care management of COVID-19 outbreak	Intensive care unit (ICU)	Review of the activities performed, and the experiences gained	National and international cooperation offers the best opportunity for survival for the critically ill.
Diaz and Dawson [20]	Development of a COVID-19 resuscitation procedure in the emergency department of paediatrics	Paediatric emergency	Discrete-event simulation	Simulation can be applied to enhance infection prevention and control initiatives, to assist developing of COVID-19 procedures, workflows, and spaces, as well as to support education teams about COVID-19 nuances.
Das [19]	Studying the influence of the COVID-19 outbreak on the current workflow	Endoscopy centre	Discrete-event simulation and Monte-Carlo analysis	Post-COVID-19 proposed workflow changes meaningfully impact productivity and operational metrics and, in turn, adversely impact financial indicators.
Tey et al. [25]	Studying the challenges of the COVID-19 pandemic	Radiation oncology	Using a modified workflow from a few cancer centres	Applied steps in the treatment of oncology patients during an infectious eruption were introduced.
Wei et al. [19]	Navigation of radiotherapy workflow and safety procedures during the COVID-19 outbreak	Radiotherapy in the cancer hospital	Review of the activities performed, and the experiences gained	Particular measures were taken to battle COVID-19 though maintaining radiotherapy care.
Yan et al. [26]	Providing recommendations for coronavirus disease 2019 prevention and infection control	Radiology	Review of the activities performed, and the experiences gained	Typical transmission-based provision, computed tomography workflow for the check-up of fever patients, as well as cleansing management of a radiology section were described.
Bettinelli et al. [2]	Providing an operative flowchart for COVID-19 patient treatment	Orthopaedic	Review of the activities performed, and the experiences gained	A workflow for patients attained in the ER in an Orthopaedic Hub is designed.

**Table 2 ijerph-17-08577-t002:** The specifications of the wards of the heart clinic.

Ward Code	Ward Name	Number of Doctors/Technicians	Number of Equipment	PDF of Service Duration (Min)	Average Service Duration (Min)
A1	Patient reception	1	-	U (5, 15)	10
A2	Electrocardiography	1	1	U (2, 15)	12
A3	Check-up	2	-	U (5, 20)	15
A4	Echocardiography	1	1	U (5, 15)	10
A5	Sport test	1	1	U (5, 60)	33
A6	Angiography	1	1	U (120, 180)	150
A7	Hospitalization	1	-	U (10, 30)	20
A8	Log out	1	-	U (10, 30)	20

PDF, probability density function.

**Table 3 ijerph-17-08577-t003:** The details of the proposed scenarios.

No.	Scenario Name	Number of Employees Added to A1	Temporary Usage of Idle Staff of Other Wards
1	Scenario 1	2	Yes
2	Scenario 2	3	Yes
3	Scenario 3	4	No

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
