# Peer review of "Simulation and Improvement of Patients’ Workflow in Heart Clinics during COVID-19 Pandemic Using Timed Coloured Petri Nets"

_ijerph, 2020, doi:10.3390/ijerph17228577_

Round 1

Reviewer 1 Report

First of all, thanks for the opportunity to review this interesting work. The objective of the work is to apply the Petri nets technique to improve the design of workflows in a heart clinic.
The topic is of interest to managers and decision-makers, especially now that the COVID-19 pandemic forced the redesign of workflows and patient access to health services to reduce the risk of nosocomial infections and prevent infection of health professionals.

Authors adequately justify their work and it seems that a good review of the literature was carried out. However, I think there are some considerations that authors should take into account to improve the quality of their work.

In fact, authors justify their work based on the COVID-19 pandemic and thus collect it at some point in the methodology (Figure 2). However, the results do not show different flows for patients with or without COVID-19. It is possible that the sequence diagram has been developed for patients who are new to the clinic, but in that case a COVID-19 screening step should be included, leading to patient isolation until results are confirmed. In addition to a new circuit in case of positive COVID-19. I think this may be the main problem with this job. There is no relationship between the justification and the work developed when including COVID-19 patients in the flow developed.

It is possible that this is reflected in Figure 6, but the quality of the image does not allow to verify it. In this regard, authors should put more effort into presenting this sequence diagram. It is one of the main results of your work. The quality of the image is quite improvable and, in addition, it includes too much information so that it can be displayed correctly at this size. Perhaps the authors could think of a web service that hosts the image and can consult it online.

Another methodological aspect to consider is sampling. How many measurements were made? How many measurements are considered necessary in this type of work according to the literature? How many did other authors do in previous studies? Were the measurements random? How and who performed the measurement?

Moreover, authors should consider that they submitted their article to a public health journal and it is compelling to clearly and understandably explain the process engineering methods used so that readers understand them and build confidence in their results. In this line, the results and data of the three proposed scenarios are not clear. The authors state that scenario 2 is directly optimal, but they do not show the data for all scenarios.

On the other hand, there are some specific aspects that should be clarified. For example:
1.- What difference do researchers establish between "Patient Reception" and "Patient Admission"? It can be confusing.
2.- Why do researchers only consider medical personnel? Presumably nurses, technicians, clerks, etc. will work in the clinic. Are these people not influencing workflows and waiting times? Furthermore, this comment is related to the generalization of the results that the authors establish at the end of the discussion (lines 335-337). This generalization is excessive. On the one hand there are the recommendations in the treatment. On the other, there are different organizational models.

Finally, authors should deepen the discussion of their results. In fact, there is no discussion in this article. An important aspect in this regard is to consider the efficiency of the proposed model.

I hope these comments are helpful.
Best regards.

Reviewer 2 Report

This study regarding patients’ waiting time in hospitals is very relevant amid COVID-19 pandemic crisis. To be published, a few major issues and concerns must be addressed:

  1. Actually, this research was targeted at a specific medical department/specialized clinic, i.e., a heart clinic. Why a heart clinic was selected and whether findings of this study could be generalizable to other medical department cases should be mentioned. Also, the title of this paper needs to reflect this limited scope of the study by including “a heart clinic” rather than “hospitals”.
  2. According to Table 1, there are lots of previous studies on the same/similar topic to this research (there are two studies even on heart clinic cases like this study). The authors of this study need to explain the main differences between this study and the prior studies and the theoretical and methodological contributions of this study beyond the previous ones.
  3. At second stage, how only three scenarios could be selected on what rationales? And, at the initial simulation, are there any starting values or default values? (If they change, what will happen to the results of this study? Have you conducted any kind of sensitivity tests regarding this issue?)    

Round 2

Reviewer 1 Report

981 / 5000  

Resultados de traducción

Thanks again for the opportunity to review this work. Authores significantly increased their quality. However, there are still some aspects of improvement: 1.- Authors must order the references so that the numbering is consecutive the first time each reference appears (1, 2, 3, ... n), avoiding the jumps that are observed in the introduction. 2.- The authors must assess whether they eliminate figures 11 and 12 or pass them on to results. No results are displayed in the discussion. The authors must contrast their results and methods with other investigations, clarify the limitations of the work, establish its strengths and propose possible lines of investigation. 3.- The discussion continues to be superficial. It is possible that the study is novel and that this makes discussion difficult, but surely these techniques were applied successfully in other areas and it is also possible to learn from them. I hope these comments are helpful. Coordial greetings,

Reviewer 2 Report

The authors have incorporated this reviewer's comments and suggestions as much as possible. 
